# Chemogenetics defines a short-chain fatty acid receptor gut–brain axis

Natasja Barki[1], Daniele Bolognini[1], Ulf Börjesson[2], Laura Jenkins[1], John Riddell[3], David I Hughes[3], Trond Ulven[4], Brian D Hudson[1], Elisabeth Rexen Ulven[4], Niek Dekker[2], Andrew B Tobin[1], Graeme Milligan[1]*

[1]Centre for Translational Pharmacology, Institute of Molecular, Cell and Systems Biology, College of Medical, Veterinary and Life Sciences, University of Glasgow, Glasgow, United Kingdom; [2]Discovery Sciences, Biopharmaceutical R&D, AstraZeneca, Gothenburg, Sweden; [3]Institute of Neuroscience and Psychology, College of Medical, Veterinary and Life Sciences, University of Glasgow, Glasgow, United Kingdom; [4]Department of Drug Design and Pharmacology, University of Copenhagen, Universitetsparken, Copenhagen, Denmark

**Abstract** Volatile small molecules, including the short-chain fatty acids (SCFAs), acetate and propionate, released by the gut microbiota from the catabolism of nondigestible starches, can act in a hormone-like fashion via specific G-protein-coupled receptors (GPCRs). The primary GPCR targets for these SCFAs are FFA2 and FFA3. Using transgenic mice in which FFA2 was replaced by an altered form called a Designer Receptor Exclusively Activated by Designer Drugs (FFA2-DREADD), but in which FFA3 is unaltered, and a newly identified FFA2-DREADD agonist 4-methoxy-3-methyl-benzoic acid (MOMBA), we demonstrate how specific functions of FFA2 and FFA3 define a SCFA–gut–brain axis. Activation of both FFA2/3 in the lumen of the gut stimulates spinal cord activity and activation of gut FFA3 directly regulates sensory afferent neuronal firing. Moreover, we demonstrate that FFA2 and FFA3 are both functionally expressed in dorsal root- and nodose ganglia where they signal through different G proteins and mechanisms to regulate cellular calcium levels. We conclude that FFA2 and FFA3, acting at distinct levels, provide an axis by which SCFAs originating from the gut microbiota can regulate central activity.

*For correspondence:
Graeme.Milligan@glasgow.ac.uk

## Editor's evaluation

This work that presents in great experimental detail how short chain fatty acids produced by gut microbiota interact with the short chain fatty acids receptors FFA2 and FFA3 significantly extends previous findings of your group in particular by identifying a new agonist for FFA2 but also because of the techniques you have used to achieve your results. In particular, that this study relies on a state of the art chemogenetics approach allows you to perform a rigorous validation of the receptor agonists you identified. You have also in your revised manuscript addressed the major concerns that were raised by the reviewers in the previous round of reviews and we believe this paper will have a significant impact in the field.

## Introduction

The gut–brain axis allows bidirectional communication between the enteric and central nervous systems. Growing evidence highlights the role that the intestinal microbiota may play in such interactions (*Bienenstock et al., 2015*) and in the development of disease (*Luca et al., 2019*). The microbiota produces a wide array of metabolites that can modulate host cells and their functions (*Chen*

*et al., 2019*; *Nicolas and Chang, 2019*). Among these short-chain fatty acids (SCFAs), particularly acetate (C2) and propionate (C3), are generated in prodigious amounts by fermentation of fiber and other nondigestible carbohydrates in the lower gut. SCFAs play central roles in homeostasis at the interface between metabolism and immunity (*Alvarez-Curto and Milligan, 2016*; *Tan et al., 2014*) and within the gut–brain axis (*van de Wouw et al., 2018*; *Dalile et al., 2019*). Although a range of their key effects are believed to be produced by activation of G protein-coupled receptors (GPCRs), both locally in the lower gut and following uptake into the systemic circulation, which of these roles are generated directly by individual receptors and via which signaling pathways and circuits remains undefined.

Over the course of evolution genes encoding many ancestral forms of GPCRs have multiplied and diversified to provide enhanced flexibility of signaling and integration of responses to either the same or closely related ligands. An example of this is within the family of receptors, FFA1, FFA2, and FFA3 that are activated by the binding of free fatty acids. Whilst FFA1 is activated by saturated and unsaturated fatty acids of chain length C10 and above both FFA2 and FFA3 are activated instead by SCFAs (*Stoddart et al., 2008b*; *Bolognini et al., 2016b*). Defining the functions of FFA2 and FFA3 is challenging due to each of overlapping potency of SCFAs at these receptors (*Hudson et al., 2012b*), a paucity of highly selective pharmacological tool compounds for FFA2 and FFA3 individually (*Milligan et al., 2017b*; *Milligan et al., 2017a*) and multiple non-GPCR-mediated effects of SCFAs (*Stoddart et al., 2008b*; *Bolognini et al., 2016b*).

To overcome these issues, and to define unambiguously specific roles of FFA2 and FFA3 in the actions of SCFAs within the gut–brain axis herein we have applied an integrated chemogenetic approach. Firstly, in addition to the use of selective FFA2 and FFA3 knockout mouse lines we have extended the use of a transgenic knockin mouse line in which we expressed a Designer Receptor Exclusively Activated by Designer Drugs (DREADD) derived from human (h) FFA2 (*Hudson et al., 2012a*) in place of endogenous mouse FFA2 (*Bolognini et al., 2019*). This variant form does not respond to the SCFAs C2 or C3 and, therefore, both in vivo and in cells and tissues derived from these mice responses to C2 or C3 cannot reflect activation of hFFA2-DREADD. Importantly, expression of FFA3 is unaffected in these animals (*Bolognini et al., 2019*) and, therefore, effects of the SCFAs may instead be mediated by FFA3. In previous studies, we identified that 4-hexadienoic acid (sorbic acid) acts as an agonist at hFFA2-DREADD but is without activity at various wild-type orthologs of FFA2 and of FFA3 (*Hudson et al., 2012a*; *Bolognini et al., 2019*). As such, effects of sorbic acid in the transgenic mice are potentially mediated by hFFA2-DREADD but not by FFA3 (*Bolognini et al., 2019*). The availability of distinct activators of any receptor is beneficial in studies to explore receptor function. Although we are unaware of any noted off-target effects of sorbic acid it has modest potency at hFFA2-DREADD and although it has extremely low toxicity (*Walker, 1990*), it is widely used as a food preservative (*Kaczmarek et al., 2019*). We wished therefore to identify additional, chemically distinct, hFFA2-DREADD agonist ligands. Here by screening libraries of small molecules with similarity to sorbic acid we identified a series of 4-methoxy-benzoic acid derivatives that also act as highly selective, and somewhat more potent activators of hFFA2-DREADD. From these we selected 4-methoxy-3-methyl-benzoic acid (MOMBA) for the studies reported herein.

Using the unique hFFA2-DREADD transgenic mice and combinations of MOMBA and C3, supported by a novel FFA3 selective activator (TUG-1907) we recently identified (*Ulven et al., 2020*), we have dissected the contributions of FFA2 and FFA3 to functions of SCFAs at levels ranging from gut transit to the activation of nodose- and dorsal root ganglion cells and communication from the colon to the spinal cord. We show that combinations of these two receptors transduce different effects of SCFAs within these pathways and via different G-protein-mediated pathways and, by so doing, integrate signals to control the microbiota–gut–brain axis.

## Results
### Identification of 4-methoxy-benzoic acid derivatives as novel hFFA2-DREADD agonists

We have previously shown that sorbic acid is a moderately potent, selective and effective agonist of hFFA2-DREADD that lacks agonist action at each of human and mouse FFA2 and both human and mouse FFA3 (*Hudson et al., 2012a*; *Bolognini et al., 2019*). However, for pharmacological studies,

particularly when exploring native cells and tissues, use of more than a single receptor-activating ligand is important to help define with certainty 'on-target' and functionally relevant responses. We, therefore, conducted a screen for novel agonists at hFFA2-DREADD using initially a receptor–β-arrestin-2 interaction assay. Here, following transient transfection of HEK293T cells to express both hFFA2-DREADD tagged at the intracellular C-terminal tail with enhanced Yellow Fluorescent Protein (hFFA2-DREADD-eYFP) and β-arrestin-2-*Renilla*-luciferase, induced proximity of eYFP and *Renilla*-luciferase allows bioluminescence resonance energy transfer (BRET) that reflects agonist-promoted interactions between the hFFA2-DREADD receptor and the arrestin (*Bolognini et al., 2019*). Using sorbic acid as a positive control, BRET signal was enhanced in a concentration-dependent manner with $pEC_{50} = 3.89 \pm 0.04$ ($n = 3$, mean ± standard deviation [SD], *Figure 1A*). We initially screened, at 100 µM, more than 1200 small molecules selected to have some structural similarity with sorbic acid and designed/collected to have good physicochemical properties and a lack of known chemical liabilities (*Figure 1—figure supplement 1A*). This provided a robust 96-well microtitre plate-based assay with calculated Z' (*Zhang et al., 1999*) routinely >0.6 (*Figure 1—figure supplement 1B*) and greater than fivefold signal to background (*Figure 1—figure supplement 1C*). Reconfirmation screens of potential hits, also conducted at 100 µM, provided positives, including compounds 565 and 1184 (*Figure 1B*). Deconvolution indicated that compound 1184 was, in fact, sorbic acid, confirming the capacity of the screen to identify effectively a previously characterized active, whilst compound 565 was MOMBA (*Figure 1B*). Potential activity of a further 320 distinct compounds, selected now on relatedness to the hits from the initial screen, was again assessed initially at a single concentration (100 µM) in 96-well format (*Figure 1C*). This resulted in the identification of further compounds including a number closely related to MOMBA, such as 4-methoxy-3-chloro-benzoic acid (compound 132) and 4-methoxy-3-hydroxy-benzoic acid (compound 235) as actives (*Figure 1C*). A number of these were at least as potent as sorbic acid with the best displaying some three- to fivefold higher potency in this assay (*Figure 1D*). Based on the common 4-methoxy-3-X-benzoic acid scaffold of MOMBA and compounds 132 and 235 that provided a defined pharmacophore, MOMBA was selected for more detailed studies and purchase of MOMBA from a separate source confirmed activity of this chemical.

## MOMBA is a highly selective agonist of hFFA2-DREADD

To be useful as hFFA2-DREADD-specific agonists compounds should not activate either wild-type human or mouse FFA2. For MOMBA this was assessed initially for wild-type hFFA2 using an equivalent BRET-based receptor–β-arrestin-2 interaction assay to that described above for hFFA2-DREADD. Following transient coexpression of hFFA2-eYFP and β-arrestin-2-*Renilla*-luciferase in HEK293 cells, whilst the SCFA propionate (C3) was an effective agonist with $pEC_{50} = 3.46 \pm 0.01$ ($n = 3$) MOMBA was without detectable activity (*Figure 1—figure supplement 2A*). For reasons that remain undefined mouse FFA2 did not allow development of an equivalent, suitably robust receptor–β-arrestin-2 interaction assay. As such, because orthologs of FFA2 can activate $G_i$-family G proteins (*Brown et al., 2003*; *Stoddart et al., 2008a*) we turned to inhibition of cAMP assays. In Flp-In T-REx 293 cells stably expressing hFFA2-eYFP, C3 inhibited forskolin-amplified cAMP levels, but neither sorbic acid nor MOMBA did so (*Figure 1—figure supplement 2B*). Equivalent results were produced in Flp-In T-REx 293 cells induced to express mouse FFA2-eYFP where again only C3 but not either sorbic acid or MOMBA was able to inhibit forskolin-amplified cAMP levels (*Figure 1—figure supplement 2B*). The closely related receptor FFA3 is also activated by SCFAs including C3 (*Brown et al., 2003*; *Stoddart et al., 2008a*). It was, therefore, vital for subsequent studies that MOMBA also lacked activity at this receptor. Using Flp-In T-REx 293 cells expressing either human or murine forms of FFA3-eYFP C3 inhibited cAMP levels whilst, again, no effect of sorbic acid or MOMBA was observed (*Figure 1—figure supplement 2C*).

The inhibition of forskolin-stimulated cAMP levels by MOMBA in cells expressing the hFFA2-DREADD receptor was concentration-dependent and MOMBA was more potent ($pEC_{50} = 5.24 \pm 0.16$, mean ± SD, $n = 3$) than sorbic acid ($pEC_{50} = 4.77 \pm 0.15$, mean ± SD, $n = 3$) (*Figure 1—figure supplement 3A*). FFA2 is also able to interact with and activate $G_q$-family G proteins as well as $G_i$-family members (*Bolognini et al., 2019*; *Brown et al., 2003*; *Stoddart et al., 2008a*). We established, therefore, that although C3 was unable to promote inositol monophosphate generation in Flp-In T-REx 293 cells expressing hFFA2-DREADD (*Bolognini et al., 2019*), MOMBA again did so in a concentration-dependent manner and with both potency and efficacy at least equivalent to sorbic

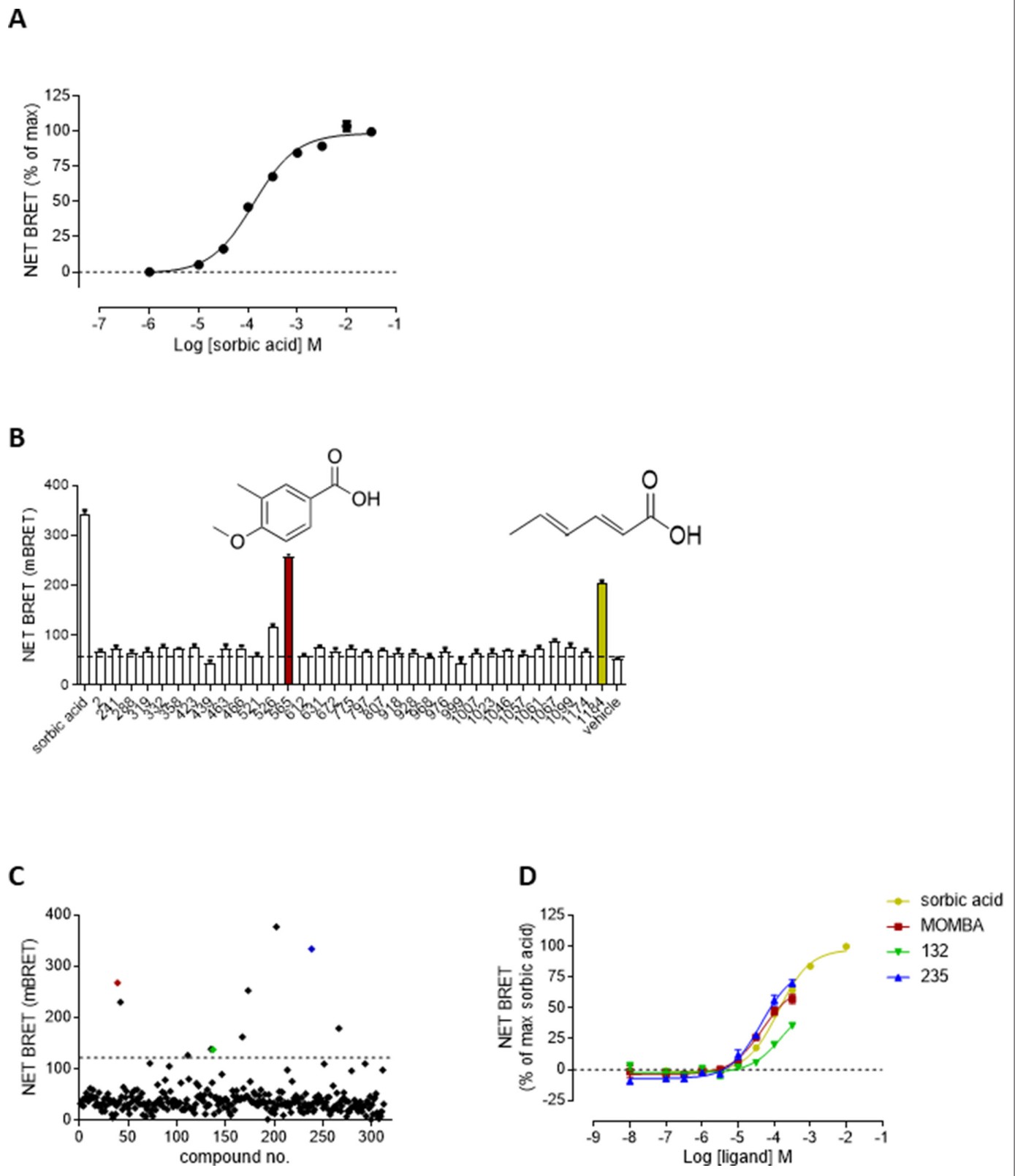

**Figure 1.** Screening for novel agonists of hFFA2-DREADD identifies 4-methoxy-3-methyl-benzoic acid (MOMBA) (A). HEK293 cells were transiently transfected to express both hFFA2-DREADD-eYFP and β-arrestin-2-*Renilla*-luciferase. Addition of sorbic acid-promoted interactions between these proteins in a concentration-dependent manner. Data are means ± standard error of the mean (SEM) of triplicates from a single experiment, representative of 6. (B) A subset of the positives from the initial screen (see *Figure 1—figure supplement 1*) were retested at 100 μM. Compounds

*Figure 1 continued on next page*

*Figure 1 continued*

565 (MOMBA) (red bar) and 1184 (sorbic acid) (yellow bar) are highlighted. Dotted line indicates basal signal. Data are from a single experiment with results plotted as mean ± SEM of triplicate assays. (C) A further 320 compounds selected on similarity to hits from *Figure 1—figure supplement 1A* were selected and screened at 100 µM in single point assays: As well as MOMBA (red) two of these were 4-methoxy-3-chloro-benzoic acid (compound 132) (green) and 4-methoxy-3-hydroxy-benzoic acid (compound 235) (blue). Dotted line indicates selection cutoff. Data are from a single experiment. (D) Concentration dependence of selected hits from C to activate hFFA2-DREADD is displayed. MOMBA (red), compound 132 (green), and compound 235 (blue). Sorbic acid (yellow) is shown as reference.

The online version of this article includes the following source data and figure supplement(s) for figure 1:

**Source data 1.** NET bioluminescence resonance energy transfer (BRET) measurements for *Figure 1*.

**Figure supplement 1.** Development of a screening assay to identify novel hFFA2-DREADD activators.

**Figure supplement 1—source data 1.** NET bioluminescence resonance energy transfer (BRET) measurements for *Figure 1—figure supplement 1*.

**Figure supplement 2.** 4-Methoxy-3-methyl-benzoic acid (MOMBA) is a selective agonist at hFFA2-DREADD.

**Figure supplement 2—source data 1.** NET bioluminescence resonance energy transfer (BRET) and percentage forskolin inhibition for *Figure 1—figure supplement 2*.

**Figure supplement 3.** 4-Methoxy-3-methyl-benzoic acid (MOMBA) is an effective and more potent orthosteric agonist of hFFA2-DREADD than sorbic acid.

**Figure supplement 3—source data 1.** cAMP levels, production of inositol monophosphates and binding of [$^{35}$S]GTPγS for *Figure 1—figure supplement 3*.

acid (*Figure 1—figure supplement 3B*). These characteristics were reiterated in measures of ligand-regulated binding of [$^{35}$S]GTPγS performed on membranes from Flp-In T-REx 293 cells expressing hFFA2-DREADD (*Figure 1—figure supplement 3C*). A key reason for using human FFA2 as the basis for the DREADD construct was that available antagonist ligands can block human FFA2, but not the mouse ortholog (*Sergeev et al., 2016*; *Sergeev et al., 2017*). The effect of an EC$_{80}$ concentration of MOMBA to stimulate [$^{35}$S]GTPγS binding was fully inhibited by increasing concentrations of both of the structurally distinct, human FFA2 ortholog specific, antagonists 4-[[[(2*R*)-1-(benzo[*b*]thien-3-ylcarbonyl)–2-methyl-2-azetidinyl]carbonyl][(3-chlorophenyl)methyl]amino]butanoic acid (GLPG0974) (pIC$_{50}$ = 7.58 ± 0.07, mean ± standard error of the mean [SEM], *n* = 3) and ((*S*)-3-(2-(3-chlorophenyl) acetamido)-4-(4-(trifluoromethyl)phenyl)butanoic acid) (CATPB) (pIC$_{50}$ = 7.63 ± 0.08, mean ± SEM, *n* = 3) (*Figure 1—figure supplement 3D*). This is consistent with, as for sorbic acid (*Bolognini et al., 2019*), MOMBA likely binding in the orthosteric pocket of hFFA2-DREADD. Further evidence that MOMBA acts as an orthosteric agonist at hFFA2-DREADD was obtained using a previously described mutation of FFA2 where Arg[180], an amino acid that co-ordinates with carboxylate-containing orthosteric agonists of FFA2, is replaced by alanine therefore eliminating orthosteric ligand function (*Stoddart et al., 2008a*). Whilst both MOMBA- and sorbic acid-stimulated hFFA2-DREADD activity (*Figure 1—figure supplement 3E*), MOMBA, as well as sorbic acid, lacked activity at the Arg[180]Ala mutant of hFFA2-DREADD (*Figure 1—figure supplement 3F*). By contrast, a previously described allosteric agonist AZ1729 (*Bolognini et al., 2016a*) that does not require Arg[180] for binding and function was equipotent at both wild type and the Arg[180]Ala mutant of hFFA2-DREADD (*Figure 1—figure supplement 3F*). These experiments established MOMBA as a useful in vitro hFFA2-DREADD-specific orthosteric agonist, but with no activity at FFA3, suggesting that MOMBA might be a favorable tool to investigate physiological roles of FFA2.

## Expression of FFA2 and FFA3 by enteric neurons of the myenteric plexus

Myenteric plexus is located between circular and longitudinal muscle and is involved in regulating smooth muscle motility. To localize the expression of FFA2 and FFA3, we used β-galactosidase reporter FFA2-knockout (FFA2-KO-βGAL) or FFA3-knockout (FFA3-KO-βGAL) mouse models. In these mice expression of β-galactosidase is driven by the relevant *Ffar2* or *Ffar3* gene promoter sequences. Fluorescent X-gal staining in myenteric plexus dissected from FFA2-KO-βGAL (21.8 ± 8.1%) (*Figure 2A*, upper panel) and FFA3-KO-βGAL mice (34.3 ± 5.6%) (*Figure 2A*, lower panel) (means ± SEM) revealed gene expression in enteric neurons as defined by staining for the neuronal RNA-binding proteins HuC/D (*Figure 2A*). β-Galactosidase staining was not found in enteric glia as defined by staining for

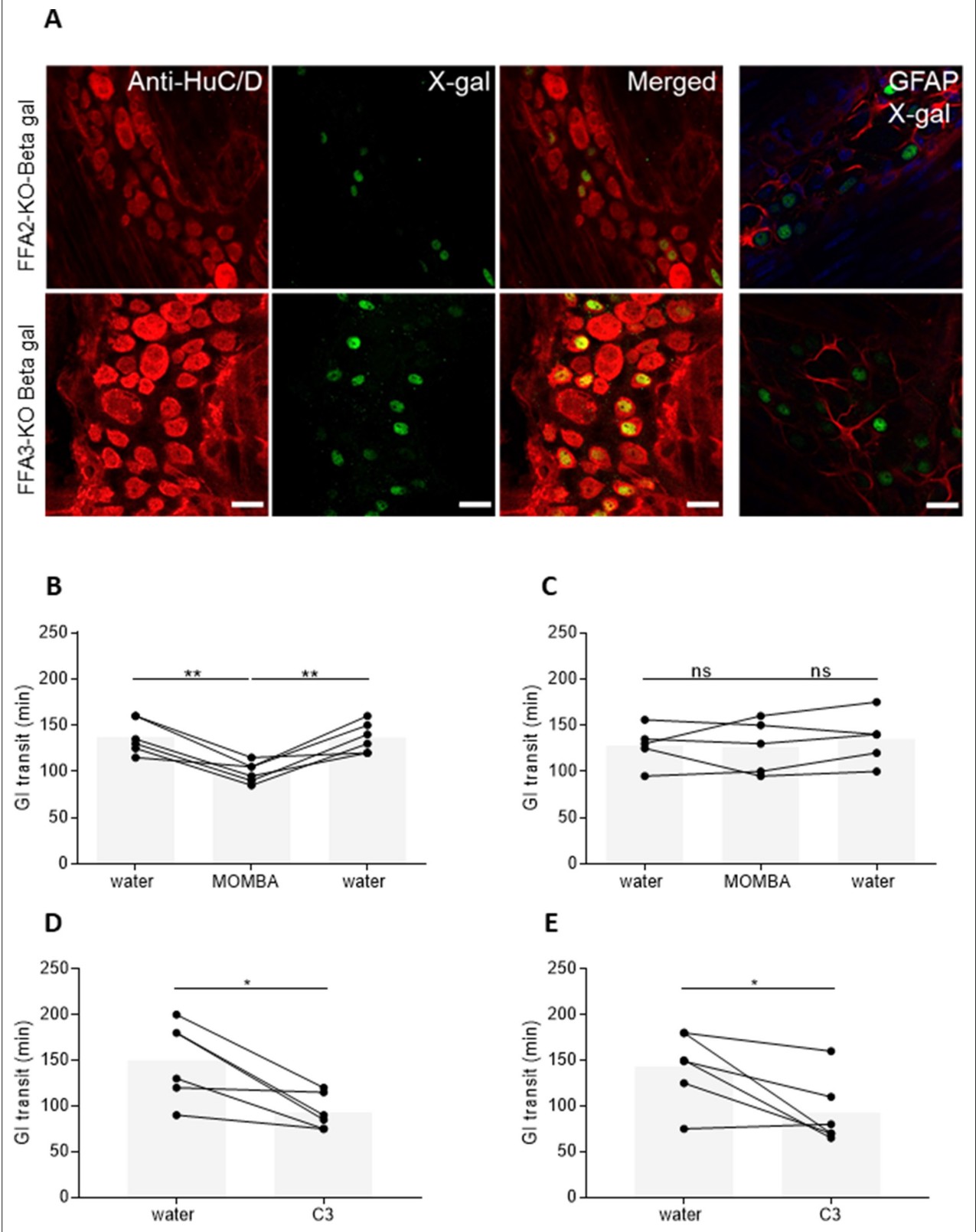

**Figure 2.** Both FFA2 and FFA3 are expressed in myenteric neurons and promote increased gut transit. (**A**) Myenteric plexus dissected from mice expressing a β-galactosidase reporter gene, driven by the *Ffar2* (upper panel) or *Ffar3* (lower panel) gene promoter sequences were immunostained with anti-HuC/D to identify enteric neurons (left-hand panels, red) and with X-gal to identify receptor-expressing cells (second panels, green). Merged images (third panels) showed coexpression. Myenteric plexus immunostained with X-gal and Glial Fibrillary Acidic Protein (GFAP) to identify enteric

*Figure 2 continued on next page*

*Figure 2 continued*

glia did not show any coexpression (right-hand panels). Blue: staining with 4',6-diamidino-2-phenylindole (DAPI) to identify cell nuclei. Scale bar = 20 µm. Male hFFA2-DREADD-HA (**B**), CRE-MINUS (**C**), FFA2-KO-βGAL (**D**), or FFA3-KO-βGAL (**E**) mice were acclimatized for 7 days with free access to drinking water. Individual animals were then gavaged with carmine red and total GI transit time measured. Following the initial transit studies, mice were provided with 4-methoxy-3-methyl-benzoic acid (MOMBA) (15 mM) or C3 (150 mM) in the drinking water as indicated. After a further 7 days GI transit of all mice was again measured. MOMBA was then removed and the mice were again provided with water followed by a further gavage with carmine red 7 days later. Data are for individual animals (*p < 0.05, **p < 0.01, ns = not significant). One-way analysis of variance followed by Bonferroni's Multiple Comparison Test.

The online version of this article includes the following source data and figure supplement(s) for figure 2:

**Source data 1.** GI transit measurements for *Figure 2*.

**Figure supplement 1.** 4-Methoxy-3-methyl-benzoic acid (MOMBA) promotes release of GLP-1 and peptide YY (PYY) in colonic preparations from hFFA2-DREADD-HA-expressing mice.

**Figure supplement 1—source data 1.** GLP-1 and peptide YY (PYY) secretion for *Figure 2—figure supplement 1*.

Glial Fibrillary Acidic Protein (GFAP) (*Figure 2A*, right-hand panels). β-Galactosidase staining equally was not observed in tissue from wild-type littermates (not shown).

## Confirmation of the ex vivo and in vivo activity of MOMBA at hFFA2-DREADD

We recently generated a 'knock-in' transgenic mouse line in which mouse FFA2 is replaced with a humanized sequence able to encode a C terminally HA-epitope-tagged form of hFFA2-DREADD (*Bolognini et al., 2019*). Upstream of the start codon we engineered a LOX-stop-LOX cassette allowing for conditional expression. In our previous experiments and herein we crossed these mice with whole body CRE-expressing mice. This resulted in the expression of hFFA2-DREADD-HA in the same tissues as mouse FFA2 in wild-type animals, and to similar levels (*Bolognini et al., 2019*), since hFFA2-DREADD is driven from the endogenous *Ffar2* promoter, and in which expression of FFA3 is unaltered (*Bolognini et al., 2019*). Control animals were mice not crossed with CRE-expressing mice and therefore did not express either wild-type FFA2 or hFFA2-DREADD but these do maintain expression of FFA3 – herein these mice are termed CRE-MINUS. We have used these mice previously in combination with the FFA2-DREADD agonist sorbic acid to define specific roles for FFA2 in gut and adipose tissue (*Bolognini et al., 2019*).

Before employing MOMBA more broadly a key requirement was to validate this hFFA2-DREADD ligand as suitable for in vivo/ex vivo studies. Entirely consistent with our previous studies with sorbic acid (*Bolognini et al., 2019*) we established that oral administration of MOMBA (15 mM) in the drinking water of hFFA2-DREADD-HA-expressing mice significantly reduced (p < 0.01) gut transit (*Figure 2B*). Importantly, subsequent removal of MOMBA from the drinking water restored transit in the same individual mice (p < 0.01) to control levels (*Figure 2B*). This clearly was an on-target hFFA2-DREADD-mediated effect because provision of MOMBA in the drinking water of CRE-Minus mice did not replicate this effect (*Figure 2C*). Furthermore, as we also observed a significant reduction (p < 0.05) in GI transit after provision of the C3 SCFA propionate in the drinking water of both FFA2-KO-βGAL (*Figure 2D*) and FFA3-KO-βGAL (*Figure 2E*) mice, this suggests that both the FFA2 and FFA3 receptors play roles in controlling muscle function.

By employing the hFFA2-DREADD/sorbic acid pairing our previous studies also described a role for FFA2 in the release of GLP-1 from enteroendocrine cells (*Bolognini et al., 2019*). Consistent with these studies, and to further define the suitability of MOMBA for subsequent ex vivo studies, we show here that MOMBA also promoted a concentration-dependent release of GLP-1 from colonic crypts prepared from hFFA2-DREADD-HA mice (*Figure 2—figure supplement 1A*). Moreover, at 100 µM MOMBA was as effective as 1 mM sorbic acid (*Figure 2—figure supplement 1A*), consistent with the higher potency of MOMBA measured in vitro (*Figure 1—figure supplement 3*). These studies also indicated that MOMBA was at least as efficacious as sorbic acid in native cells and tissues. In control experiments C3 (10 mM) was unable to induce release of GLP-1 from hFFA2-DREADD-HA colonic crypts as expected (*Bolognini et al., 2019*), eliminating a role for FFA3. To examine the role of FFA2 in GLP-1 release further the colons from hFFA2-DREADD-HA mice were dissected and mounted in the chamber of an organ bath in which ligands could be perfused through the tissue. In this preparation

MOMBA caused a rapid and sustained increase in the release of GLP-1 (*Figure 2—figure supplement 1B*) in a manner that was completely inhibited by the hFFA2-specific antagonist CATPB (*Bolognini et al., 2019*; *Sergeev et al., 2016*) a ligand that also antagonizes the hFFA2-DREADD (see *Bolognini et al., 2019* and *Figure 1—figure supplement 3D*). The CRE-MINUS form of these mice is functionally akin to a FFA2 knockout line (*Bolognini et al., 2019*). In colonic tissue from such CRE-MINUS animals MOMBA was unable to promote release of GLP-1 (*Figure 2—figure supplement 1C*), further confirming this endpoint to reflect activation of hFFA2-DREADD rather than any potential 'off-target', nonreceptor, mediated effect of MOMBA.

Together these studies demonstrated that the hFFA2-DREADD/MOMBA pairing was able to give equivalent outcomes as previously described for the hFFA2-DREADD/sorbic acid pairing both ex vivo and in vivo but, importantly, at lower ligand concentration. Furthermore, these studies established MOMBA as a valid tool compound to probe novel FFA2 physiology.

## Colonic release of peptide YY is also mediated specifically by FFA2

Peptide YY (PYY) is expressed in similar subsets of enteroendocrine 'L-cells' as GLP-1 (*Nohr et al., 2013*; *Lu et al., 2018*) and perfusion of colonic tubes isolated from hFFA2-DREADD-HA mice with MOMBA also resulted in a rapid, markedly enhanced (p < 0.01), but in this case transient, release of PYY (*Figure 2—figure supplement 1D*). This effect of MOMBA was not observed in the presence of CATPB (*Figure 2—figure supplement 1D*) and was also not produced in equivalent preparations generated from CRE-MINUS animals (*Figure 2—figure supplement 1E*). A distinct feature of the hFFA2-DREADD-HA-expressing mice is that the appended C-terminal HA epitope tag allows exquisite immunochemical detection of cells expressing the receptor (*Bolognini et al., 2019*). Consistent with the MOMBA-induced release of PYY, costaining of colonic sections from the hFFA2-DREADD-HA mice with both anti-HA and anti-PYY antibodies identified a subset of cells that as well as being positive for PYY also expressed hFFA2-DREADD-HA (*Figure 2—figure supplement 1F*). Such colocalized staining was lacking in tissue sections isolated from wild-type animals, although identification of PYY-expressing cells was equivalent (*Figure 2—figure supplement 1F*).

## Activation of FFA3 in the proximal colon promotes firing of vagal afferents

It has been established in mouse that enteroendocrine cells can synapse with vagal neurons (*Kupari et al., 2019*). We next wished to assess whether SCFAs were able to stimulate afferent nerve activity in the proximal colon (innervated by both vagal and spinal afferents). We therefore recorded nerve activity from such afferents (*Figure 3A, B*) whilst perfusing the inside of proximal colon isolated from wild-type mice. Perfusion of tissue with C3 resulted in a marked increase (p < 0.01) in rate of nerve firing in this preparation (*Figure 3B, C*). C3 produced a similar effect (p < 0.05) in tissue isolated from hFFA2-DREADD-HA-expressing mice (*Figure 3C*) potentially indicating a key role for FFA3. A potential contribution of FFA2 to such effects was also assessed by perfusing tissue from hFFA2-DREADD-HA-expressing mice with MOMBA. However, no statistically significant effect of MOMBA was observed (*Figure 3D*) and this was also the case when MOMBA was applied to tissue taken from the CRE-MINUS animals (*Figure 3D*). This combination of studies indicates that FFA2 does not contribute significantly to the observed effects of C3 in wild-type mice but would be consistent with a specific role for FFA3. To confirm the role of FFA3, the recently described FFA3 selective activator TUG-1907 (*Ulven et al., 2020*) was perfused into colon isolated from wild-type or FFA3-KO-βGAL mice (*Figure 3E*). TUG-1907 increased nerve activity in tissue from C57/BL/6 mice (p < 0.01), but this was absent in tissue from FFA3-KO-βGAL mice, confirming the role of FFA3 in increasing peripheral nerve activity from the proximal colon in response to SCFAs.

## Cells of the nodose ganglia express FFA2

Sensory signaling from the proximal colon is communicated to the nodose ganglia (NDG) via the vagal nerve. Anti-HA staining of NDG isolated from hFFA2-DREADD-HA mice showed clear immunoreactivity that overlapped to only a limited degree with the neuronal marker PGP9.5 (7.5 ± 3.8% [mean ± SEM]; *Figure 4—figure supplement 1A*) and was also noted in various cells located between the neurons (*Figure 4—figure supplement 1A*). Anti-HA staining was specific for hFFA2-DREADD-HA expression because no such staining was observed in wild-type mice (not shown). We next wanted

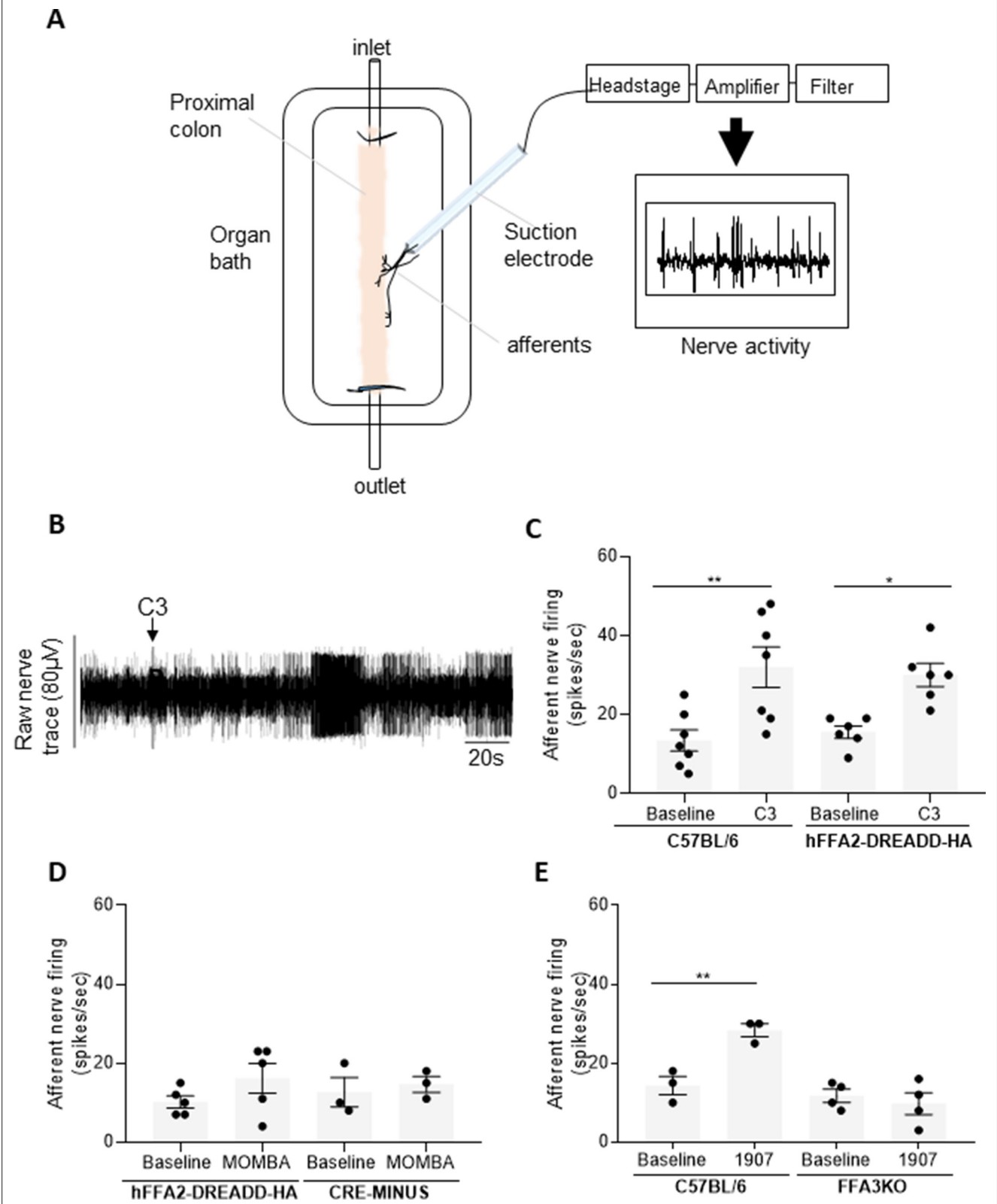

**Figure 3.** Colonic SCFAs promote firing of afferents via FFA3 (**A**). A schematic illustration of the ex vivo proximal colon preparation. The proximal colon is superfused in the recording chamber and is cannulated at both ends. Intraluminal infusion is achieved using a syringe pump (100 µl/min). A nerve branch is dissected and inserted into a suction electrode and recording is made using neurolog and Spike software. (**B**) A representative trace of the colonic afferent nerve signal counting individual spikes above a preset threshold (spikes/s). Introduction of C3 is highlighted. (**C**) The ability of

*Figure 3 continued on next page*

*Figure 3 continued*

C3 to promote afferent nerve activity was compared to buffer (baseline) in segments of the proximal colon taken from either wild-type C57BL/6 or hFFA2-DREADD-HA-expressing mice. C3 increased nerve firing in both preparations (**p < 0.01 and *p < 0.05, one-way analysis of variance followed by Bonferroni's Multiple Comparison Test). (D) Similar studies were performed with 4-methoxy-3-methyl-benzoic acid (MOMBA) using tissue from either hFFA2-DREADD-HA or CRE-MINUS mice. No significant effect of MOMBA was detected (one-way analysis of variance followed by Bonferroni's Multiple Comparison Test). (E) TUG-1907 (3 μM) was able to increase nerve activity in tissue from C57BL/6 but not in tissue taken from FFA3-KO-βGAL mice (**p < 0.01, one-way analysis of variance followed by Bonferroni's Multiple Comparison Test). *Figure 3—source data 1*.

The online version of this article includes the following source data for figure 3:

**Source data 1.** Multiunit nerve activity for *Figure 3*.

to determine if these receptors were functionally coupled to G-protein signaling. Hence, cells from NDG were isolated from hFFA2-DREADD-HA mice and following dispersal and plating onto matrigel-coated coverslips were loaded with the $Ca^{2+}$ indicator dye Fura-8-AM. Single-cell $Ca^{2+}$ imaging indicated that a proportion of these cells (37.2 ± 8.4% [mean ± SEM]) responded to addition of MOMBA (*Figure 4—figure supplement 1B*) whereas a markedly larger (p < 0.001) proportion (86.9 ± 6.5%, mean± SEM) responded to C3 (*Figure 4—figure supplement 1B*). The effect of MOMBA clearly reflected activation of hFFA2-DREADD-HA because, once more, this was fully blocked by coaddition of the hFFA2-specific antagonist CATPB (*Figure 4—figure supplement 1B*). Moreover, in equivalent cells isolated from wild-type mice no response to MOMBA was recorded (1.6 ± 1.0%, mean ± SEM) (*Figure 4—figure supplement 1C*), whilst a similar proportion of the cells (85.0 ± 5.5%, mean ± SEM) of the cells responded to C3. These data indicate that functional hFFA2-DREADD-HA and FFA3 receptors are both present on cells of NDG but do not assess directly to what degree they may be coexpressed nor on which of the complex and diverse makeup of cells that comprise the NDG (*Kupari et al., 2019*) are these receptors expressed.

## Dorsal root ganglia express both FFA2 and FFA3

Whilst expression of FFA2 in cells of the NDG suggests a role in nutrient sensing, in addition to that uncovered previously for FFA3 in sympathetic ganglia (*Kimura et al., 2011*), we wished to also define possible functional expression and coexpression of receptors for SCFAs in other sensory ganglia. We therefore turned to dorsal root ganglia (DRG). The expression of FFA3 in DRG of mouse has previously been noted by using a FFA3-Red Fluorescent Protein (RFP) reporter mouse line in which expression of RFP, instead of the FFA3 receptor, is driven by *Ffar3* gene promoter sequences (*Nohr et al., 2015*). Limited information on potential expression of FFA2 in such ganglia is available. We examined this in colonic innervating DRGs that had been isolated from the T9-L2 region of the spinal cord of hFFA2-DREADD-HA-expressing mice (*Figure 4A*). Anti-HA staining was specific for hFFA2-DREADD-HA expression because no such staining was observed in wild-type mice (*Figure 4A*) whereas neuronal staining with anti-PGP9.5 was equivalent (*Figure 4A*). As with NDG, whilst a degree of overlap of anti-HA immunostaining was observed with that for the neuronal marker PGP9.5 (9.3 ± 5.8%, mean ± SEM), which was both widely distributed and detected in tissue from both hFFA2-DREADD-HA-expressing (*Figure 4A*) and CRE-MINUS mice, the most intense anti-HA staining was in cells that were interspersed between the PGP9.5-positive neurons (*Figure 4A*). Consistent with HA staining, a similar pattern of expression was also observed with X-gal staining corresponding to FFA2 promoter function in tissue from FFA2-KO-βGAL mice (*Figure 4B*, left panel). Furthermore, promoter function and hence expression of FFA3 (*Figure 4B*, middle panel) was also identified in DRGs by X-gal staining in tissue from FFA3-KO-βGAL mice. By contrast X-gal staining was absent in wild-type littermates (*Figure 4B*, right panel).

## Elevation of $Ca^{2+}$ in isolated dorsal root ganglion cells defines functional roles for both FFA2 and FFA3 receptors

To assess function of FFA2, cells from DRGs taken from hFFA2-DREADD-HA-expressing mice were prepared as for those from NDG. Addition of MOMBA produced a rapid and partially sustained increase in intracellular $[Ca^{2+}]$ (*Figure 4—figure supplement 1D*) in a substantial proportion (41.4 ± 10.0%, mean ± SEM) of the cells tested (*Figure 4C* and *Supplementary file 1*). By contrast addition of C3-promoted intracellular $[Ca^{2+}]$ elevation in most (82.1 ± 3.8%, mean ± SEM) cells (*Figure 4D* and *Supplementary file 2*) and this was generally more robust than effects of MOMBA (*Figure 4—figure*

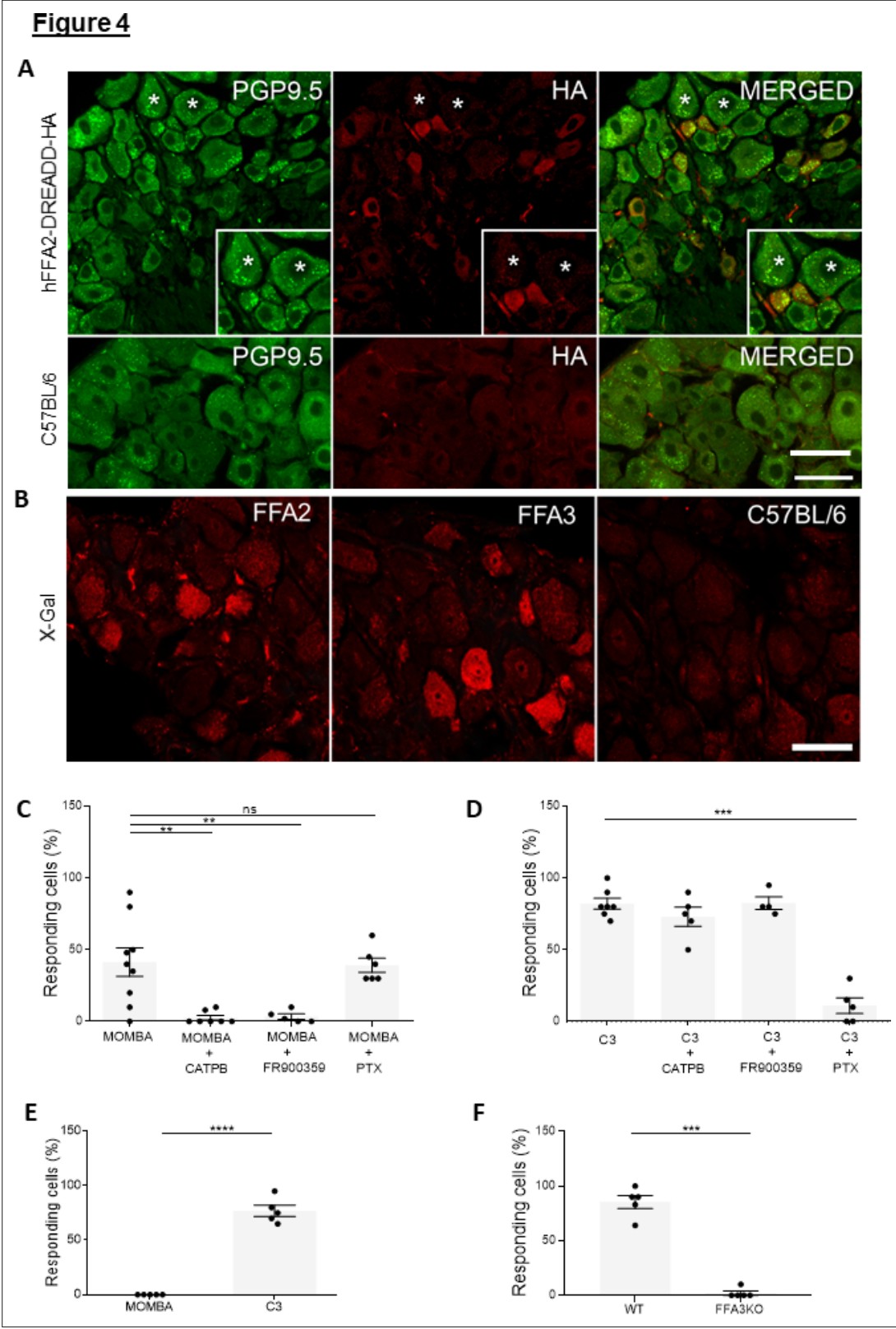

**Figure 4.** Both FFA2-DREADD-HA and FFA3 are expressed and functional in cells of dorsal root ganglia: the two G-protein-coupled receptors (GPCRs) elevate $Ca^{2+}$ by different mechanisms (A). Sections of dorsal root ganglia taken from hFFA2-DREADD-HA-expressing mice were immunostained with anti-PGP9.5 to identify neurons (left-hand panel, green) and with anti-HA to detect the receptor (middle panel, red). Merging of such images (right-hand panel) showed modest coexpression (see main text for quantification) but with additional anti-HA staining interspersed between neurons. Inserts:

*Figure 4 continued on next page*

*Figure 4 continued*

focus on regions in which individual cells express PGP9.5 but not HA (star), express HA reactivity but not PGP9.5 (chevron), or coexpress both PGP9.5 and hFFA2-DREADD-HA (double arrow). (B) Fluorescent X-gal staining (red) of dorsal root ganglia (DRG) sections isolated from mice expressing β-galactosidase reporter gene, which is driven either by the *Ffar2* (left-hand panel) or *Ffar3* gene promoters (middle panel). DRG sections from wild-type mice were also immunostained with X-gal (right-hand panel). Scale bar = 20 μm. (C, D) Single-cell $Ca^{2+}$ imaging studies were performed on cells isolated from DRGs taken from hFFA2-DREADD-HA-expressing mice. Cells were exposed to 4-methoxy-3-methyl-benzoic acid (MOMBA) (C) or C3 (D). In various examples cells were pre-treated CATPB, with the Gq/G11 inhibitor FR900359 (15 min) or pertussis toxin (24 hr) prior to addition of agonist. CATPB blocked the effect of MOMBA (**p < 0.01) but not C3. FR900359 also blocked the effect of MOMBA (**p < 0.01) but not C3, whilst pertussis toxin treatment blocked the effect of C3 (***p < 0.001) but not MOMBA. One-way analysis of variance followed by Bonferroni's Multiple Comparison Test. Pertussis toxin treatment blocked the effect of C3 (***p < 0.001) but not MOMBA. One-way analysis of variance followed by Bonferroni's Multiple Comparison Test. (E) Cells dispersed from DRGs isolated from CRE-MINUS mice were used to assess the ability of ligands to elevate $Ca^{2+}$. No effect of MOMBA was recorded whilst C3 was effective in most of the cells tested (****p <0.0001, unpaired *t*-test). (F) TUG-1907 (3 μM) was able to elevate $Ca^{2+}$ levels in DRG cells from wild type but not FFA3-knockout mice (***p < 0.001, unpaired *t*-test).

The online version of this article includes the following source data and figure supplement(s) for figure 4:

**Source data 1.** Intracellular calcium data (expressed as relative fluorescence) for *Figure 4*.

**Figure supplement 1.** FFA2 is expressed and functional (intracellular calcium response) in nodose ganglia of mice.

**Figure supplement 1—source data 1.** Calcium imaging values for *Figure 4—figure supplement 1*.

supplement 1D). Initial stimulation by MOMBA did not prevent subsequent elevation of $[Ca^{2+}]$ in the same cells in response to C3 following washout of MOMBA (*Figure 4—figure supplement 1D*). This indicates both that at least a proportion of these cells coexpress FFA2 and FFA3 and that exposure to MOMBA did not desensitize cells to subsequent exposure to C3. This is consistent with MOMBA and C3 producing elevation of $[Ca^{2+}]$ via distinct receptors. Once more the effect of MOMBA was clearly an on-target effect of the ligand at hFFA2-DREADD-HA because responses to MOMBA were absent in cells derived from CRE-MINUS control mice and when cells isolated from hFFA2-DREADD-HA mice were preincubated with the hFFA2 receptor-specific antagonist CATPB (*Figure 4C* and *Supplementary file 1*). By contrast, effects of C3 on cells isolated from hFFA2-DREADD-HA mice were not blocked by CATPB (*Figure 4D* and *Supplementary file 2*) and the proportion of cells that responded to C3 was not different in DRGs isolated from CRE-MINUS animals (*Figure 4E*). Together these data support the notion that in DRG cells isolated from hFFA2-DREADD-HA mice MOMBA stimulates $Ca^{2+}$ mobilization via FFA2-DREADD-HA and C3 stimulates $Ca^{2+}$ responses via FFA3. To further support the conclusion that responses to C3 in DRG-derived cells from hFFA2-DREADD-HA-expressing mice were indeed mediated via FFA3 we again employed the novel FFA3 activator TUG-1907 (*Ulven et al., 2020*). This ligand produced substantial elevation of $Ca^{2+}$ in isolated DRG-derived cells (*Figure 4F*). As this ligand did not produce such effects in DRG-derived cells isolated from FFA3 knockout mice (*Figure 4F*), this confirmed the presence and functionality of FFA3 in hFFA2-DREADD-HA-expressing mice.

## Distinct mechanisms of FFA2- and FFA3-mediated Ca²⁺ elevation in cells of DRG

With the higher number of cells that could be isolated from DRGs compared to NDG we were able to explore mechanisms underlying the observed elevation of $Ca^{2+}$. The ability of MOMBA to promote elevation of $Ca^{2+}$ levels in DRG-derived cells taken from hFFA2-DREADD-HA mice was blocked by coincubation with the $G\alpha_q/G\alpha_{11}$ inhibitor FR900359 (*Schrage et al., 2015*; *Figure 4C* and *Supplementary file 1*). This is consistent with the cation being released from inositol 1,4,5 trisphosphate ($IP_3$)-sensitive internal stores. By contrast effects of MOMBA were unaffected by pretreatment with pertussis toxin (*Figure 4C* and *Supplementary file 1*) which causes ADP-ribosylation of $G\alpha_i$ family subunits. Very differently, responses to C3 were instead blocked by pretreatment with pertussis toxin, but not by treatment with FR900359 (*Figure 4D* and *Supplementary file 2*). This is also consistent with the effect of C3 being mediated by FFA3 in these cells, as FFA3 is known to selectively activate $G_i$ G proteins (*Brown et al., 2003*; *Stoddart et al., 2008a*), and potentially proceeding via release of β/γ complexes from such $G_i\alpha$-containing G-protein heterotrimers (*Smrcka and Fisher, 2019*; *Senarath et al., 2018*). These results indicate that activation of each of FFA2 and FFA3 results in elevation of $Ca^{2+}$ in DRG-derived cells, but that the two receptors engage with different G proteins and use

distinct mechanisms to mediate these effects. This is also entirely consistent with the lack of cross-desensitization of the ligands noted earlier.

## Gut SCFA receptors promote activation of neurons in the dorsal horn of the spinal cord

DRG afferents innervate the dorsal horn of the spinal cord. As such we also wished to assess whether SCFAs present in the gut might promote central nervous system stimulation and, if so, if this was mediated by FFA2 and/or FFA3. To do so, initially we perfused either saline or C3 via the rectum into the colon of anaesthetized wild-type mice. Following sacrifice 2 hr later activation of neurons in the dorsal horn of the spinal cord was assessed by measuring induction of expression of the early immediate gene c-Fos (*Coggeshall, 2005*; *Boyle et al., 2019*). C3 produced a substantial increase in c-Fos-positive neurons compared to saline (p < 0.05) (*Figure 5A*, compare *Figure 5B, C*, *Supplementary file 3*). To identify a potential contribution of FFA2 to this effect we performed equivalent experiments in hFFA2-DREADD-HA-expressing mice. Here, introduction of MOMBA also produced a significant increase (p < 0.05) in the number of c-Fos-positive neurons compared to saline (*Figure 5A*, compare *Figure 5D, E*, *Supplementary file 3*), although this was less extensive than produced by C3 in the wild-type animals. By contrast MOMBA did not promote such an effect in the CRE-MINUS animals (*Figure 5A*, compare *Figure 5F, G*), confirming this effect of MOMBA to reflect activation of the hFFA2-DREADD-HA receptor and that activation in the colon of both of FFA2 and likely FFA3 results in spinal activation. To further confirm the contribution of FFA3, we perfused C3 into the colorectal region of hFFA2-DREADD-HA mice. This also resulted in a significant increase (p < 0.05) in c-Fos-positive neurons.

5-Hydroxytryptamine (5-HT) is an important neurotransmitter in various GI functions, including secretion, motility, and visceral hypersensitivity. More than 90% of the total body 5-HT is found within the GI tract, primarily in enteroendocrine cells, enterochromaffin cells, and within neurons. While many of the 5-HT receptors play a role in gut physiology, the ligand gated 5-HT$_3$ receptor is mainly involved in signaling from sensory vagal afferents. To assess whether 5-HT$_3$ also plays a role in gut–spinal interaction induced by activation of SCFA receptors, the 5-HT$_3$ receptor antagonist granisetron (1 µM) was administered prior to the administration of either MOMBA or C3. Preincubation with granisteron did not inhibit spinal c-Fos expression promoted by either MOMBA or C3 (*Figure 5—figure supplement 1*, suggesting therefore, that the 5-HT$_3$ receptor does not play a key role in this process).

## Discussion

Here, we establish and validate the existence of a SCFA–gut–brain axis by which activation of either of the SCFA receptors FFA2 and FFA3 in the colon results in changes in spinal cord activity. In this way, we demonstrate a process whereby SCFAs generated in high amounts by gut microbiota-mediated fermentation of fiber can regulate central activity. Our study also begins to address the mechanism(s) by which this axis operates and the individual contributions of the two receptors. Firstly, we determine that gut sensory afferent neuronal firing is selectively enhanced by activation of FFA3 in the gut. Secondly, that functional FFA2 and FFA3 receptors, coupling via G$_q$ and G$_i$ G-protein subfamilies, respectively, are expressed by both neuronal and non-neuronal cells in DRG and NDGs and that gut-generated SCFAs can regulate spinal cord activity via both FFA2 and FFA3 receptors present in the gut, and at the level of extraspinal sensory ganglia (e.g., DRG/NDG).

These conclusions stem from our application and extension of the concept of DREADD receptors. Although most studies apply this technology in the context of the muscarinic acetylcholine family of GPCRs (*Armbruster et al., 2007*; *Urban and Roth, 2015*; *Wess et al., 2013*; *Bradley et al., 2018*) we have instead used this to engineer a DREADD of the FFA2 receptor. Previously, we demonstrated that by introduction of two amino acid alterations the hFFA2 receptor loses binding and responsiveness to SCFAs, but in parallel gains responsiveness to sorbic acid, thereby creating an FFA2-DREADD with an associated DREADD ligand (*Hudson et al., 2012a*). By generating a knockin mouse line expressing an epitope-tagged hFFA2-DREADD-HA in place of the wild-type mouse ortholog we engineered a physiological system whereby in both ex vivo and in vivo settings FFA2-mediated responses are activated directly by sorbic acid whilst FFA3 responses are potentially identified as those responding to SCFAs because now FFA3, but not the FFA2-DREADD, still responds to SCFAs. Using these animals

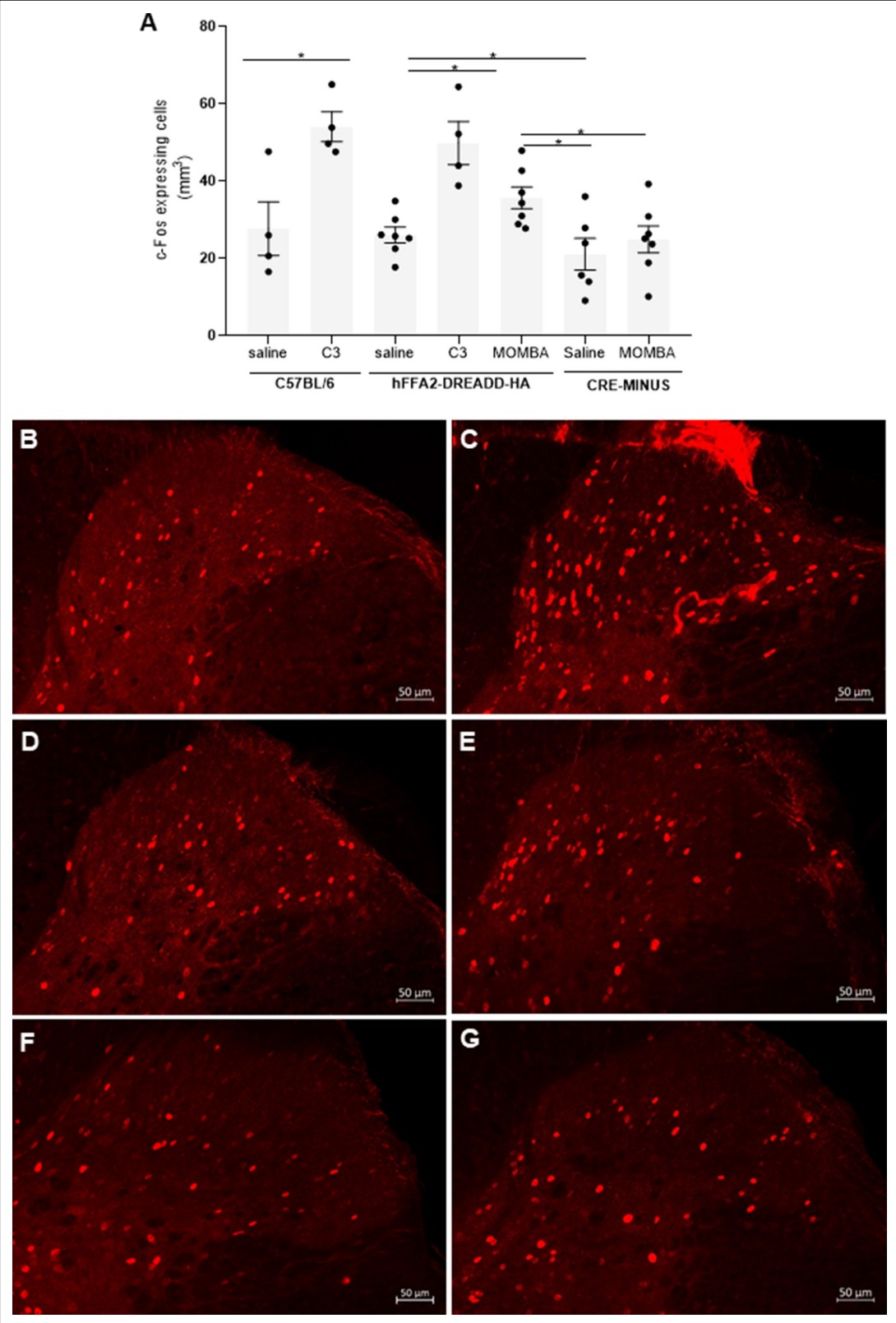

**Figure 5.** Gut short-chain fatty acid receptors promote activation of neurons in the dorsal horn of the spinal cord: roles for both FFA2 and FFA3 (A). 2 hr after introduction of saline, C3 or 4-methoxy-3-methyl-benzoic acid (MOMBA) into the colon of wild-type C57BL/6, hFFA2-DREADD-HA, or CRE-MINUS mice the number of c-Fos-expressing cells in the dorsal horn of the spinal cord was quantified. Data points represent individual animals. C3 increased c-Fos-expressing cells in wild-type (*p < 0.05) and hFFA2-DREADD-HA-expressing mice, MOMBA also did so in hFFA2-DREADD-HA expressing (*p <

*Figure 5 continued on next page*

*Figure 5 continued*

0.05) but not CRE-MINUS mice (unpaired *t*-test). Representative images of dorsal horn from C57BL/6 plus saline (**B**), or plus C3 (**C**); hFFA2-DREADD-HA plus saline (**D**), or plus MOMBA (**E**); CRE-MINUS plus saline (**F**), or plus MOMBA (**G**) treatments are shown for illustration. Scale bar = 50 µm.

The online version of this article includes the following source data and figure supplement(s) for figure 5:

**Source data 1.** c-Fos-expressing cells per mm³ for *Figure 5*.

**Figure supplement 1.** Spinal c-Fos expression induced by activation of short-chain fatty acid (SCFA) receptors.

we have previously identified unique and specific roles for FFA2 in enteroendocrine hormone release and adipose tissue function, as well as in gut transit (*Bolognini et al., 2019*).

Despite the success of these earlier studies and that sorbic acid is known to have extremely low toxicity (*Walker, 1990*), we wished to identify other, chemically distinct, hFFA2-DREADD agonist ligands. Here, we describe the identification and validation of the alternative FFA2-DREADD agonist MOMBA, which displays improved potency and equivalent efficacy to sorbic acid. In validation studies, we demonstrated that MOMBA mirrored the results previously described for sorbic acid in FFA2-DREADD-HA-expressing animals and therefore MOMBA was taken forward to investigate novel physiological roles for FFA2/FFA3. The first of these new findings established the expression of hFFA2-DREADD-HA in enteroendocrine cells coexpressing the hormone PYY, and that specific activation of the FFA2-DREADD results in PYY release. Whilst benzoic acids and some of their salts are employed in food preservation it is clearly a marked advantage to now have two structurally distinct agonists that act as activators of the hFFA2-DREADD receptor, not least because the probability that they would have the same 'off-target' effects is low.

Through combinations of FFA2 and FFA3 KO β-galactosidase reporter mouse lines alongside immunological identification of the HA epitope tag sequence built into the FFA2-DREADD-HA construct, we have been able to define expression patterns in mouse of FFA2 and FFA3 throughout the gut–brain axis. Moreover, by application of the FFA2-DREADD-HA/MOMBA pairing and our recent development and characterization of the most potent FFA3 selective activators yet described (*Ulven et al., 2020*), we also here define that NDG and DRG each express functional FFA2 and FFA3. Although, selective activation of either FFA2 or FFA3 results in elevation of Ca$^{2+}$ in cells of these ganglia it is remarkable that, whilst both these GPCRs normally are activated by the same group of SCFAs, their effects on intracellular Ca$^{2+}$ levels are produced by completely different mechanisms. The significance of these different mechanisms in terms of kinetics of response, post-transcriptional modifications and even transcriptional regulation, and hence plasticity of the ganglia and spinal cord neurons, in integration of functions will be topics of future analysis.

It is well appreciated that FFA3 couples highly selectively to members of the pertussis toxin-sensitive G$_i$ family of G proteins (*Brown et al., 2003*; *Stoddart et al., 2008a*). As such, it was anticipated that in cells and tissues from hFFA2-DREADD-HA-expressing mice FFA3-mediated effects should be blocked by pretreatment with pertussis toxin. An established mechanism to allow G$_i$-coupled receptors to elevate Ca$^{2+}$ is via β/γ complexes released by activation of G$_i$α-containing heterotrimers (*Smrcka and Fisher, 2019*; *Senarath et al., 2018*). By contrast FFA2, and indeed the hFFA2-DREADD, in transfected cell lines is able to promote activation and downstream signaling via a wide range of G proteins including G$_{q/11}$ and G$_i$ proteins (*Hudson et al., 2012a*; *Bolognini et al., 2019*; *Brown et al., 2003*; *Stoddart et al., 2008a*). However, in DRG, we can conclude that FFA2 couples to Ca$^{2+}$ signaling solely via G$_{q/11}$ because effects of MOMBA were fully attenuated when cells had been pretreated with the highly selective G$_{q/11}$ inhibitor FR900359 (*Schrage et al., 2015*). Whilst release of the hormone GLP-1 also requires selective activation of G$_{q/11}$ family G proteins (*Bolognini et al., 2019*), by contrast, our previous studies established that FFA2-mediated regulation of lipolysis is transduced by G$_i$-family G proteins (*Bolognini et al., 2019*). Hence, in transfected cells although FFA2 shows the capacity to couple to each of G$_{q/11}$ and G$_i$ and other G proteins (*Bolognini et al., 2019*), in physiological cell types the receptor appears to couple selectively to either G$_i$ (adipose tissue) or G$_{q/11}$ (DRG and enteroendocrine cells). How FFA2 selects between coexpressed G proteins in physiologically relevant cell types remain unclear but will also be a topic for future analysis.

Perhaps most intriguing amongst the outcomes reported herein is the capacity for SCFA receptors that are activated by agonists introduced into the colon to cause activation of cells at the level of the dorsal horn of the spinal cord. These observations support a role for gut microbiota-derived SCFAs

in directly co-ordinating and controlling responses between the gut and the central nervous system. Given the rapidly emerging evidence that the microbiota and their dysbiosis can be key contributors to a variety of long-term central nervous system deficits and malfunctions (*Ma et al., 2019*; *Wang et al., 2018*; *Cussotto et al., 2018*; *Rieder et al., 2017*), the current studies provide insights into direct chemical links via a SCFA–gut–brain axis. The fact that we can manipulate this axis via synthetic small molecule chemical ligands (e.g. MOMBA and TUG-1907) suggests that pharmacologically targeting FFA2 or FFA3 might provide a therapeutic strategy for the treatment for various CNS disorders (*Silva et al., 2020*). Future studies will therefore also examine this directly.

## Methods

### Compounds

Compounds for screening were selected in two stages from the AstraZeneca compound collection. First, a larger set of carboxylic acids was assembled to generate an initial list of actives by screening. The first set of compounds aimed to balance (1) similarity to known agonists (including sorbic acid), (2) chemical diversity, and (3) physicochemical properties consistent with favorable ADME (absorption, distribution, metabolism, and excretion) profiles of molecules. To this end, carboxylic acids were identified by substructure matching, and compounds with either (1) molecular weight <350 Da, (2) calculated octanol–water partition coefficient log p > 3.5, (3) calculated polar surface area >110 Å$^2$, or (4) unwanted chemical groups were removed. After clustering, a diverse subset of 1210 compounds was manually selected. The second stage of compound selection involved identifying close analogs to the active molecules in the first screen. Thus, 320 compounds were selected using multiple substructure searches. Subsequent batches of MOMBA were purchased from FLUOROCHEM, Hadfield, UK.

### Primary compound screening

Compounds were assayed in a single point BRET assay at 100 μM containing 1% (vol/vol) Dimethyl-sulphoxide (DMSO). Basal wells containing assay buffer with 1% DMSO, and positive stimulation wells containing 1 mM sorbic acid were included in all plates. Each plate contained two '*Renilla* luciferase' points containing cells lacking receptor expression. Data from primary screening were analyzed using Microsoft Excel software and activities of the compounds were calculated using the following formula:

Activity (%) = (mBRET compound − mBRET basal)/(mBRET stim − mBRET basal) × 100, where mBRET compound is the mBRET value obtained from wells treated with the test compound, mBRET basal is the average of the mBRET values obtained from wells treated with assay buffer and mBRET stim is the average of the mBRET values obtained from cells treated with 1 mM sorbic acid. Compounds were considered as possible hits if the activity was higher than the mean + 3 × SD of the overall activity in the whole assay. Hits were considered confirmed if the activity remained over this threshold in a second independent assay. Reliability of the assay was estimated by calculating $Z'$ values (*Zhang et al., 1999*) for each plate, using the formula: $Z'$ = 1 - {[(3 × σstim) + (3 × σbasal)]/(μstim − μbasal)} where σstim and σbasal are the SD values of wells containing 1 mM sorbic acid and assay buffer, respectively and μstim and μbasal are the means for wells containing 1 mM sorbate and assay buffer, respectively.

### Animal maintenance

The generation and characterization of both transgenic FFA2-DREADD-HA-expressing and CRE-MINUS mouse lines are detailed in *Bolognini et al., 2019*. All animals were bred as homozygous onto a C57BL/6N background. FFA2-KO-βGAL and FFA3-KO-βGAL mice were provided by AstraZeneca. Male and female mice were used in this study unless otherwise stated. Mice were fed ad libitum with a standard mouse chow diet. Maintenance and killing of mice followed principles of good laboratory practice in accordance with UK national laws and regulations. All experiments were conducted under a home office licence held by the authors.

### Cell lines

HEK-293T cells were maintained in Dulbecco's modified Eagle's medium (DMEM) supplemented with 0.292 g/l L-glutamine, penicillin/streptomycin mixture and 10% (vol/vol) fetal bovine serum (FBS) at 37°C in a 5% $CO_2$ humidified atmosphere. For experiments using transiently transfected HEK-293T

cells, transfections were carried out using 1 mg/ml polyethyleneimine (PEI) (MW-25000) and experiments conducted 48 hr post-transfection.

Flp-In T-REx-293 cells (Invitrogen) were maintained in DMEM without sodium pyruvate, supplemented with 10% (vol/vol) FBS, 1% penicillin/streptomycin mixture, and 10 µg/ml blasticidin at 37°C in a 5% $CO_2$ humidified atmosphere. To generate Flp-In T-REx 293 cells able to inducibly express the various FFA receptor constructs, cells were cotransfected with a mixture containing the desired cDNA in pcDNA5/FRT/TO vector and pOG44 vector (1:9) using PEI. Transfected cells were selected using 200 µg/ml hygromycin B. Expression of the appropriate construct from the Flp-In T-REx locus was induced by treatment with 100 ng/ml doxycycline for 24 h.

None of the cells used in this study have been tested for microplasma and the identity of the cells lines were authenticated using STR DNA profiling.

## Cell signaling assays

Each of regulation of forskolin-amplified levels of cAMP and production of inositol monophosphates (IP$_1$) was performed as detailed in *Bolognini et al., 2019*. Binding of [$^{35}$S]GTPγS is detailed in *Sergeev et al., 2016*.

## β-Arrestin-2 recruitment assays

β-Arrestin-2 recruitment to a receptor of interest was assessed using a BRET-based assay. HEK-293T cells were cotransfected with the desired receptor tagged with eYFP at its C terminus, and with β-arrestin-2-*Renilla* luciferase (ratio 4:1), using 1 mg/ml PEI, linear MW-25000 (ratio 1:6 DNA/PEI). Subsequent experimental procedures were performed as described previously (*Bolognini et al., 2019*). The final concentration of coelenterazine-h used as substrate was 5 µM. BRET measurements were performed using a PHERAstar FS reader (BMG-Labtech, Offenburg, Germany). The BRET ratio was calculated as emission at 530 nm/emission at 485 nm. Net BRET was defined as the 530/485 nm ratio of cells coexpressing *Renilla* luciferase and eYFP minus the BRET ratio of cells expressing only the *Renilla* luciferase construct in the same experiment. This value was multiplied by 1000 to obtain mBRET units.

## Afferent nerve recordings

Nerve activity was recorded as previously described by *Nullens et al., 2016*. Following cervical dislocation, the distal colon with the cecum was immediately removed. The colon was placed in a recording chamber that was continually perfused with oxygenated (95% $O_2$ and 5% $CO_2$) Krebs-bicarbonate solution (composition, mM: NaCl 118.4, NaHCO$_3$ 24.9, CaCl$_2$ 1.9, MgSO$_4$ 1.2, KH$_2$PO$_4$ 1.2, glucose 11.7) at 35°C (pH 7.4). The colon was visualized using a dissection microscope to enable identification of the nerve bundles from which afferent recordings were to be made. These nerve bundles were carefully dissected into individual branches. One nerve branch was inserted into a recording electrode (tip diameter 50–100 µm) attached to neurology headstage (NL100, Digitimer Ltd, UK), AC amplifier (NL104), and filter (NL125, band pass 300–4000 Hz) and captured by a computer via a Power 1401 interface and Spike2 software (version 5.14, Cambridge Electronic Design, UK). Multiunit nerve recordings were performed at baseline. The preparation was allowed to stabilize for 30 min, before starting the protocol. Following stabilization of afferent nerve recording, drugs were applied intraluminally into the colon. Afferent responses to drug application were compared to a 30-min control.

## Immunohistochemistry

Colonic tissues were isolated, mounted, and processed using previously described (*Bolognini et al., 2019*) methods. Sections were immunostained with anti-HA (1:100; Sigma-Aldrich anti-HA high-affinity clone 3F10) and anti-PYY (1:400 Abcam-Ab1 ab22663) antibodies and mounted with VECTASHIELD Vibrance Antifade Mounting Medium with DAPI (Vector Laboratories). Images were taken with an EVOS M7000 Imaging System (Thermo Fisher Scientific).

## DRG and NDG: cell isolation and calcium imaging

Colonic innervating DRGs were isolated from the T9-L2 region of the spinal cord of wild-type and transgenic animals and immediately placed in cold Hanks' balanced salt solution (HBSS; Sigma-Aldrich). Isolated DRGs were initially digested with HBSS containing L-cysteine (0.3 mg/ml) and papain (2.0

mg/ml) for 20 min at 37°C. The solution was removed and replaced with HBSS contain collagenase (4.0 mg/ml) and dispase (4.0 mg/ml) (20 min at 37°C) for further digestion. The collagenase solution was then replaced with DMEM to stop the reaction. The DRGs were finally dissociated by mechanical trituration using a pipette. Dissociated cells were plated on matrigel-coated coverslips and placed in an incubator (37°C and 5% $CO_2$). Following a 2-hr incubation cells were flooded with 90% DMEM (Sigma) supplemented with 10% fetal calf serum and 1% PenStrep and further incubated overnight at 37°C and 5% $CO_2$.

To measure intracellular calcium and its potential regulation, dissociated cells on the coverslips were loaded with Fura 8-AM (3 μM) (Stratech Scientific Limited) for 20 min at 37°C in the dark. Coverslips were then placed in a recording chamber and mounted onto a invert fluorescent microscope (Nikon TE2000-E; Nikon Instruments, Melville, NY) equipped with a (NA = 1.3) oil-immersion Super Fluor objective lens (×40), an Optoscan monochromator (Cairn Research, Faversham, Kent, UK) and a digital Cool Snap-HQ CCD camera (Roper Scientific/Photometrics, Tucson, AZ). Illumination of the preparation was achieved by a Meta Fluor imaging software (Molecular Devices, San Jose CA, version 7.8.8).

Clusters of cells were randomly selected for real-time imaging and continuously perfused with (4-(2-hydroxyethyl)-1)-piperazineethanesulphonic acid (HEPES) buffer (composition: HEPES 10 mM, NaCl 135 mM, glucose 10 mM, KCl 5 mM, $CaCl_2$ 2 mM, and $MgCl_2$ 1 mM, pH 7.4) for 20 min at room temperature. All test ligands were diluted in HEPES buffer and perfused through the chamber for 3 min, followed by a final application of the $Ca^{2+}$ ionophore ionomycin (5 μM), as a positive control.

Results are expressed as relative fluorescence (RF), $n$ numbers are presented as $N$ = number of mice and $n$ = number of cells. Similar methods were used for cells from NDG.

## Detection of c-Fos expression in spinal cord

Transverse spinal cord sections (40 μm thick) from T9 to T10 segments were cut on a vibrating blade microtome Leica VT1200 or VT1000S (Leica). Free-floating sections were incubated in 0.03% $H_2O_2$ in PBS for 30 min, and in 50% ethanol for a further 30 min. Sections were then incubated with goat anti-c-Fos primary antibody (diluted 1:500; Santa Cruz Biotechnology; RRID: AB_2629503) for 72 hr at 4°C, followed by overnight incubation in biotinylated secondary antibody (1:500, Jackson Laboratory). Immunolabeling was visualized using a tyramide signal amplification tetramethylrhodamine kit (NEL702001KT, Perkin Elmer), as described previously (*Hughes et al., 2013*). Sections were costained with NeuN (diluted 1:500) and mounted on glass slides in Vectashield anti-fade mounting medium with DAPI.

## Colon stimulation

Colonic stimulation was performed by gently inserting a blunt, lubricated catheter a distance of 2.5 cm from the anus. Vehicle or test compounds (200 μl, at room temperature) were administered at a steady rate over 1 min by pressure injection. Animals were allowed to recover and monitored for discomfort/pain for 2 hr, prior to transcardial perfusion of 10 ml of Ringer solution followed by 500 ml of 4% depolymerized formaldehyde in a 0.1 M phosphate buffer under terminal anaesthesia.

## Cryosectioning and immunostaining

Following cervical dislocation, DRGs were quickly dissected out and fixed in 4% PFA for 90 min. The specimens were cryosectioned at 30 μm and thaw-mounted onto adhesive slides (Leica). Slides were washed in Tris-buffered saline (TBS) containing 0.3% Triton X-100 and incubated with blocking buffer (TBS, 0.3% Triton X-100, 3% goat serum, 5% Bovine Serum Albumin [(BSA)]) for 2 hr, followed by incubation with primary antibodies (rat-HA, 1:100 rabbit-PGP9.5, 1:1000, rabbit-X-gal, 1:250) for 24 hr (4°C). Slides were further incubated for 2 hr (RTP) with secondary antibody 1:400 (Alexa 448 fluor goat anti-rat, or Alexa fluor 647 goat anti-rabbit). Slides were mounted with Vectashield mounting solution. Fluorescent images were visualized and captured with a Zeiss confocal microscope.

## In vivo gastrointestinal transit

GI transit was measured as previously described by *Bolognini et al., 2019*. Briefly, a cohort of male (12–18 weeks) hFFA2-DREADD-HA, wild-type, CRE-MINUS, FFA2-KO-βGAL, and FFA3-KO-βGAL mice were single caged with free access to food and water. After one week of acclimatization, mice

were gavaged with a solution of carmine red (300 µl; 6%; Sigma-Aldrich) suspended in methylcellulose (0.5%; Sigma-Aldrich). Total GI transit time was measured as the time between oral gavage (time = 0) and the appearance of the first red pellet. Following the initial transit studies, half the mice from each group were randomly selected and provided with MOMBA (15 mM) in the drinking water, whereas the other half (control) continued drinking water without MOMBA. After 1 week the GI transit of all mice was again measured. In certain studies, MOMBA was then removed and the mice were again provided with water followed by a further gavage with carmine red a week later.

## Colonic crypt isolation

As previously described (*Bolognini et al., 2019*) the colon was immediately removed and placed in ice-cold Krebs solution. The colon was cut longitudinally and pinned on a sylgard-coated dish. The muscle was gently removed and the remaining tissue was chopped using a scalpel. The tissue was then washed three times with cold PBS. For tissue digestion, the colon was placed in medium containing 0.3 mg/ml collagenase XI (Sigma-Aldrich) for 15 min at 37°C. The supernatant was then collected and the remaining tissue was further digested with collagenase. This process was repeated two more times to allow complete digestion of the colon. Isolated crypts were plated on matrigel (Corning)-coated wells and incubated overnight at 37°C and 5% $CO_2$ in DMEM (25 mM glucose) supplemented with 10% FBS, 1% glutamine, and 1% penicillin/streptomycin. On the following day, all wells were washed with 138 buffer and challenged with test compounds. After 2 hr, the supernatants and lysates were centrifuged (4°C, 18,000 × $g$). Active GLP-1 secretion was measured by ELISA (Millipore).

## Organ bath studies

GLP-1 and PYY secretion from intact colon was investigated in hFFA2-DREADD-HA, CRE-MINUS, and wild-type mice as previously described in detail (*Bolognini et al., 2019*). The entire colon was removed and placed in an purpose built organ bath (3 ml) perfused with carbogenated (95% $O_2$–5% $CO_2$) Krebs solution (composition, mM: NaCl 120; KCl 5.9; $NaH_2PO_4$ 1.2; $MgSO_4$ 1.2; $NaHCO_3$ 15.4; $CaCl_2$ 2.5; glucose 11.5) (34°C). The colon was attached from either end to an inlet and outlet port, allowing intraluminal perfusion (10 ml/hr) of vehicle or test ligands using a syringe pump (Sigma-Aldrich).

GLP-1 and PYY secretion was assessed by perfusing either Krebs or MOMBA (0.1 mM) through the lumen. To further assess the specificity of MOMBA, MOMBA was also applied in the presence of the human FFA2-specific antagonist CATPB (10 µM). In this case, CATPB was also applied 15 min before the coapplication of MOMBA (0.1 mM) and CATPB (10 µM). During intraluminal perfusion, supernatants were collected every 5 min. Total PYY (Phoenix) and active GLP-1 (Millipore) concentration was measured by ELISA.

## Data analysis and curve fitting

All data in this manuscript represent the mean ± SEM or SD where noted of at least three independent experiments. Data analysis and curve fitting were conducted using the GraphPad Prism software package 8.1.0 (GraphPad). Concentration–response data were fit to three-parameter sigmoidal concentration–response curves. Statistical analyses were performed using a two-tailed $t$-test, one- or two-way analysis of variance analyses followed by Dunnett or Bonferroni post hoc test as indicated. Allosteric parameters were calculated by using the operational model equation as described previously (*Bolognini et al., 2019*).

## Acknowledgements

The studies reported were funded by the Biotechnology and Biosciences Research Council (grant numbers BB/L027887/1 [to GM], BB/L02781X/1 [to ABT], and BB/S000453/1 [to GM and ABT and BDH]) and from Lundbeck Foundation (R181-2014-3247, R307-2018-2950) to ERU. We thank Nidhal Selmi at AstraZeneca for scientific input during screening and SAR expansion resulting in identification of MOMBA. We thank Prof LA Blackshaw (Queen Mary University of London) for training in extracellular nerve recording of proximal colon and access to equipment and facilities. We acknowledge the BSU facilities at the Cancer Research UK Beatson Institute (C596/A17196) and the Biological Services at the University of Glasgow.

# Additional information

### Competing interests
Ulf Börjesson, Niek Dekker: Is an employee of Astra-Zeneca. The author declares that no other competing interests exist. The other authors declare that no competing interests exist.

### Funding

| Funder | Grant reference number | Author |
|---|---|---|
| Biotechnology and Biological Sciences Research Council | BB/L027887/1 | Graeme Milligan |
| Biotechnology and Biological Sciences Research Council | BB/S000453/1 | Andrew Tobin |

The funders had no role in study design, data collection, and interpretation, or the decision to submit the work for publication.

### Author contributions
Natasja Barki, Conceptualization, Data curation, Investigation, Methodology, Validation, Writing – original draft; Daniele Bolognini, Formal analysis, Investigation, Methodology; Ulf Börjesson, Trond Ulven, Elisabeth Rexen Ulven, Niek Dekker, Investigation, Resources; Laura Jenkins, Formal analysis, Investigation, Methodology, Validation; John Riddell, Supervision; David I Hughes, Investigation, Methodology, Supervision; Brian D Hudson, Methodology; Andrew B Tobin, Conceptualization, Funding acquisition, Methodology, Resources, Supervision, Validation, Visualization, Writing – original draft, Writing – review and editing; Graeme Milligan, Conceptualization, Data curation, Funding acquisition, Methodology, Project administration, Resources, Supervision, Validation, Visualization, Writing – original draft, Writing – review and editing

### Author ORCIDs
David I Hughes ⬤ http://orcid.org/0000-0003-1260-3362
Trond Ulven ⬤ http://orcid.org/0000-0002-8135-1755
Andrew B Tobin ⬤ http://orcid.org/0000-0002-1807-3123
Graeme Milligan ⬤ http://orcid.org/0000-0002-6946-3519

### Decision letter and Author response
Decision letter https://doi.org/10.7554/eLife.73777.sa1
Author response https://doi.org/10.7554/eLife.73777.sa2

# Additional files

### Supplementary files
• Supplementary file 1. 4-Methoxy-3-methyl-benzoic acid (MOMBA) promotes $Ca^{2+}$ elevation in subsets of dorsal root ganglia (DRG)-derived cells in a FFA2 and $G_q/G_{11}$-dependent manner. *n* cells isolated from DRGs from N hFFA2-DREADD-HA-expressing mice were assessed for capacity to elevate intracellular $[Ca^{2+}]$ in response to MOMBA or MOMBA following the indicated treatments. See also *Figure 4B*.

• Supplementary file 2. C3 promotes $Ca^{2+}$ elevation in subsets of DRG-derived cells in a non-FFA2 and $G_i$-dependent manner. *n* cells isolated from dorsal root ganglia (DRGs) from N hFFA2-DREADD-HA-expressing mice were assessed for capacity to elevate intracellular $[Ca^{2+}]$ in response to C3 or C3 following the indicated treatments. See *Figure 4C* for details.

• Supplementary file 3. Activation of gut short-chain fatty acid (SCFA) receptors promotes c-Fos expression in the dorsal horn of the spinal cord.

• Transparent reporting form

## Data availability

All data generated or analysed during this study are included in the manuscript or are available from the authors.

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
