## [Editor Report]

This work that presents in great experimental detail how short chain fatty acids produced by gut microbiota interact with the short chain fatty acids receptors FFA2 and FFA3 significantly extends previous findings of your group in particular by identifying a new agonist for FFA2 but also because of the techniques you have used to achieve your results. In particular, that this study relies on a state of the art chemogenetics approach allows you to perform a rigorous validation of the receptor agonists you identified. You have also in your revised manuscript addressed the major concerns that were raised by the reviewers in the previous round of reviews and we believe this paper will have a significant impact in the field.

---

## [Decision Letter]

**Decision letter after peer review:**

Thank you for submitting your article "Chemogenetics defines a short chain fatty acid receptor gut-brain axis" for consideration by *eLife*. Your article has been reviewed by 2 peer reviewers, and the evaluation has been overseen by a Reviewing Editor and Mone Zaidi as the Senior Editor. The reviewers have opted to remain anonymous.

As you will see while reviewer 2 is overall positive, reviewer 1 raises a number of issues regarding the manuscript. Those concerns are:

– The use of of a whole-body Cre mouse which may call into questions the conclusions of the paper.

– Another concern the lack of Cre-minus controls in various experiments.

– The reviewer also noted that there is not a strong rationale presented in the manuscript for the focus on the DRG.

Given the extent of the concerns raised by reviewer 1 I am sorry to say that the paper is not acceptable for publication in its present form. We would however, be happy to consider a revised version of the paper provided it addresses ALL the concerns of reviewer 1 and 2. Those are deemed essential revisions by the editors.

We hope you will find the reviews constructive.

*Reviewer #1 (Recommendations for the authors):*

1. Why are wild type C57BL/6 mice used as controls in the BOMBA motility studies of Figure 2? A more appropriate control would be the CRE-MINUS mice that do not express hFFA2-DREADD. For the motility studies, these animals are essentially FFA2 KO, and MOMBA should not activate FFA3. Do these animals show no effect on motility after BOMBA treatment?

2. There is a similar problem with the DRG studies of Figure 4. On page 13, CRE-MINUS mice are mentioned in reference to Figure 4, but the Figure itself is labeled C57BL/6, and on page 14, wild type littermates are mentioned. For the DRG studies, studying CRE-MINUS mice would provide powerful evidence that expression of the knockin receptor is not leaky, but is it not clear that this was done.

3. In the c-Fos activation studies of Figure 5, we are provided data with CRE-MINUS mice, but only in the setting of MOMBA. No results are shown for saline or C3. Why this omission?

*Reviewer #2 (Recommendations for the authors):*

This is a very solid and convincing study. The manuscript is well written and the data compelling support the authors' notion of a SCFA-gut-brain axis. This will considerably extend understanding of microbiota-produced SCFA actions in the body. Although the manuscript has many strengths, I am puzzled by the need for screens of chemical libraries to identify new agonists for the designer FFA transgene. Possibly some text needs to be added to allow for appreciation of this point.

---

## [Author Response]

Reviewer #1 (Recommendations for the authors):1. Why are wild type C57BL/6 mice used as controls in the BOMBA motility studies of Figure 2? A more appropriate control would be the CRE-MINUS mice that do not express hFFA2-DREADD. For the motility studies, these animals are essentially FFA2 KO, and MOMBA should not activate FFA3. Do these animals show no effect on motility after BOMBA treatment?

We have actually performed essentially all the control studies with ‘Creminus’ animals as well as with wild type and the various knock-out mouse lines but excluded some data in an attempt to make the manuscript less dense. We have now included these important controls as requested by the reviewer (page 29, Figure 2C).

2. There is a similar problem with the DRG studies of Figure 4. On page 13, CRE-MINUS mice are mentioned in reference to Figure 4, but the Figure itself is labeled C57BL/6, and on page 14, wild type littermates are mentioned. For the DRG studies, studying CRE-MINUS mice would provide powerful evidence that expression of the knockin receptor is not leaky, but is it not clear that this was done.

This is important data and was included in the original manuscript but in the supplementary data. To address the reviewers concerns and make the control more visible we have moved the control data from supplementary into the main manuscript. This now appears on page 31, Figure 4E.

3. In the c-Fos activation studies of Figure 5, we are provided data with CRE-MINUS mice, but only in the setting of MOMBA. No results are shown for saline or C3. Why this omission?

To address the reviewers concerns we have added an important control – that of saline in CRE-minus mice this now gives the control for any off-target activity of MOMBA, which not detected (page 32, Figures 5A and 5F).

Reviewer #2 (Recommendations for the authors):This is a very solid and convincing study. The manuscript is well written and the data compelling support the authors' notion of a SCFA-gut-brain axis. This will considerably extend understanding of microbiota-produced SCFA actions in the body. Although the manuscript has many strengths, I am puzzled by the need for screens of chemical libraries to identify new agonists for the designer FFA transgene. Possibly some text needs to be added to allow for appreciation of this point.

We thank the reviewer for their support – we point the reviewer to the answer in the public review regarding the screen.